# Age-appropriate vaccination and associated factors among children aged 12–35 months in Ethiopia: A multi-level analysis

Bekelu Teka Worku[1], Eshetu Alemayehu Wordofa[2], Gadisa Senbeto[3], Beakal Zinab[4], Ebissa Bayana Kebede[ID][5]*, Fira Abamecha[6], Gurmessa Tura Debela[1], Negalign Birhanu[7], Yibeltal Siraneh[7], Dessalegn Tamiru[4]

1 Department of Population and Family Health, Institute of Health, Jimma University, Jimma, Ethiopia, 2 Department Epidemiology, Institute of Health, Jimma University, Jimma, Ethiopia, 3 Department Environmental Health, Institute of Health, Jimma University, Jimma, Ethiopia, 4 Department Nutrition and Dietetics, Institute of Health, Jimma University, Jimma, Ethiopia, 5 School of Nursing, Institute of Health, Jimma University, Jimma, Ethiopia, 6 Department of Health Behavior and Society, Institute of Health, Jimma University, Jimma, Ethiopia, 7 Department of Health Policy and Management, Institute of Health, Jimma University and Ethiopia, Jimma, Ethiopia

* ebisabayana@ymail.com

**Data Availability Statement:** The data used for this study is included within the article itself.

**Funding:** The author(s) received no specific funding for this work.

## Abstract

### Background

Age-appropriate vaccination is a crucial public health measure to prevent morbidity and mortality worldwide. Despite its importance, there has been insufficient emphasis on tackling this problem. Therefore, this study aimed to determine the prevalence of age-appropriate vaccination and associated factors in Ethiopia.

### Method

Data from 1077 children aged 12-35months were extracted from the Ethiopian Mini Demographic and Health Survey 2019 using a prepared data extraction checklist and included in the analysis. The extracted data was analyzed using STATA version 14.0. Descriptive and inferential statistics were applied, followed by multilevel logistic regression. Significant variables were identified at p-value < 0.05 within 95% confidence level and AOR.

### Result

The pooled prevalence of age-appropriate vaccination in this study was 21.17% (95%CI: 18.73–23.61). Factors like mothers age > = 40 years (AOR = 4.05, 95%CI: 1.03, 15.83), 35–39 years (AOR = 4.62, 95%CI: 1.27,16.71), 25–29 years (AOR = 4.07, 95%CI: 1.18,14.03), Maternal secondary education (AOR = 1.85, 95% CI: 1.06, 3.22), Maternal primary education (AOR = 1.60, 95% CI: 1.07, 2.41) and rural residence (AOR = 0.34, 95%CI: 0.23,0.51) were significant predictors of age-appropriate vaccinations.

**Competing interests:** The authors have declared that no competing interests exist

## Conclusion

This study concluded that the prevalence of age-appropriate vaccination of children in Ethiopia is below the desired level. Hence, the stakeholders should prioritize timely vaccination of children, similar to as the efforts made to increase overall coverage.

## Introduction

Vaccination involves exposing children to a weakened or killed version of a pathogen, provoking an immune response without causing serious infections [1]. Immunization is a crucial public health intervention that can help to reduce child morbidity and mortality rates and increase life expectancy [2]. Common vaccine-preventable diseases among children include measles, diphtheria, influenza, tetanus, pertussis, hepatitis, mumps, pneumonia, polio, and rotavirus [3, 4].

To combat these vaccine-preventable diseases, the Expanded Program on Immunization (EPI) was launched by the World Health Organization (WHO) in 1974, aiming to provide immunization to all children worldwide. In 1984, the WHO standardized a vaccination schedule for six vaccines: Bacillus Calmette-Guérin (BCG), Diphtheria-Tetanus-Pertussis (DTP), Oral Polio, and Measles [5, 6]. Similarly, Ethiopia launched EPI in 1980, aiming for a 100% immunization rate within 10 years, though the target remained unachieved even after 20 years [7, 8].

Ensuring that children receive age-appropriate vaccinations is crucial to safeguarding them from vaccine-preventable diseases (VPD) [9]. Universal vaccination through routine and catch-up schedules is essential for quality healthcare, linked to improved health outcomes, cognitive development, productivity, and cost savings [10–12]. The World Immunization Agenda 2030 emphasizes increasing equitable access to vaccines to leave no one behind [13]. Despite these efforts, the burden of vaccine-preventable disease remains a global challenge.

Currently, full vaccination of children at appropriate age is a global public problem, with vaccination coverage falling back or stagnating [14]. Timely vaccination rates among children ranges from 38% to 65% in the Philippines and less than 50% Sub-Saharan African countries [15, 16]. In Ethiopia, age-appropriate vaccination practice is below half in several regions, with the Afar region (43.7%), Oromia (33.7%), and North East Ethiopia (39.1%) [17–19]. However, the pooled full vaccination coverage in Ethiopia is greater than 65%, indicating poor age-appropriate vaccination in different regions [20]. Additionally, the magnitude of missed opportunities for routine vaccination is high (39.8%) in Ethiopia [21].

Despite remarkable improvements in child health services, age-appropriate childhood immunization is still challenging in Ethiopia [22]. To address this Ethiopia developed strategies and policies like the Health Sector Transformation Plan (HSTP), reaching every district (RED) and sustainable outreach service (SOS) approaches and comprehensive multiyear immunization plans (cMYP) [7, 23]. Most vaccinations are age-specific, and some vaccinations are not recommended after a certain age, making compliance with age-appropriate vaccinations crucial for maximum effectiveness against vaccine-preventable diseases [9].

Revealing information on the age appropriateness of vaccination from national data provides a more complete view of vaccination coverage in the country. The information is essential for clarifying the status of missing and delayed vaccinations, which stakeholders, including policymakers can use to address the gaps in immunization programs. Although some studies have assessed age-appropriate vaccination in Ethiopia, none have used the most recent

national data. Therefore, this study aimed to conduct a multi-level analysis to determine the level of age-appropriate vaccination status and associated factors in Ethiopia using the 2019 Ethiopia Mini Demographic and Health Survey [24].

## Materials and methods

### Source of data and study design

The study was conducted in Ethiopia, the largest nation in the Horn of Africa. Ethiopia is the second most populous country in Africa and ranks 11th in the world populous country. According to the United Nations Children's Fund, there are 13 million under five years of age children in Ethiopia [25]. The study was done from January-February 2024. This study used secondary data from the Ethiopia Mini Demographic Survey of 2019 (EMDHS). The EMDHS was a cross-sectional study conducted on identified areas defined as enumeration areas constituting geographic areas. It is national representative data that was collected through a community-based cross-sectional survey from March 21, 2019, to June 28, 2019. The 2019 EMDHS is the second EMDHS and the fifth DHS implemented in Ethiopia. The Ethiopian Public Health Institute [24] conducted the survey in collaboration with the Central Statistical Agency and the Federal Ministry of Health, with technical assistance from ICF and support from development partners [24].

### Population

This study focused on the children aged 12–35 months, sampled from all child vaccination-related data in Ethiopia. Children with at least one confirmed vaccination history and whose vaccination card was seen at home or from health facilities were included. Children were excluded if their vaccination dates didn't align with their birth dates.

### Sample size determination and sampling techniques

The study included 1077 children aged 12–35 months who were vaccinated according to EPI guidelines, with data available in the 2019 EMDHS. The EMDHS utilized the census frame from the 2019 Ethiopia Population and Housing Census (EPHC), established by the CSA, consisting of 149,093 census enumeration areas (EAs). Each EA, on average, covers 131 households and contains data regarding EA location, urban or rural residency, and estimated residential household count. Ethiopia's administrative structure during the survey comprised of nine geographical regions and two administrative cities [24].

The sample design aimed at generating key indicators for the entire country, urban and rural areas individually, and for each of the nine regions and the two administrative cities. The sample selection process occurred in two stages within 21 strata, stratifying each region into urban and rural areas. In the first stage, 305 EAs (93 urban and 212 rural) were chosen with a probability proportional to EA size. In the second stage, involved selecting a fixed number of 30 households per cluster were selected using equal probability systematic selection from a newly established household listing. The survey conducted between March 21, 2019, and June 28, 2019, interviewed 8,855 women aged 15–49 from a nationally representative sample of 8,663 households, aiming to provide estimates at national, regional, urban, and rural levels [24].

**Operational definitions.** *Age-appropriate Vaccination*. Vaccination of children timely within the WHO-recommended window period for basic vaccines: BCG (birth– 8 weeks), Penta1 (DPT-HepB-Hib) and OPV1 (6 weeks– 14 weeks); Penta2 (DPT-HepB-Hib), and OPV2 (10 weeks– 18 weeks); Penta3(DPT-HepB-Hib), and OPV3 (14 weeks– 24 weeks) and

measles vaccine (9 months– 11 months) [26]. Vaccination was coded as '1' if received within the recommended period and '0' if before or after the recommended period. Age-appropriate vaccination was calculated from the summation of all "yes" scores for each specific basic vaccine type considered for this study.

According to the EMDHS report, the child must receive at least: one dose of BCG vaccine, which protects against tuberculosis; three doses of DPT-HepB-Hib, which protects against diphtheria, pertussis (whooping cough), tetanus, hepatitis B, and Haemophilus influenzae type b; three doses of polio vaccine; and one dose of measles vaccine [24]. Therefore, we considered these vaccines in the analysis.

### Data extraction tools and quality control methods

The data extraction checklist was adapted from the EMDHS to extract necessary data, including socio-demographics, maternal and obstetric factors, and information-related factors. Before analysis, data was extracted depending on the inclusion criteria, and variables were identified, recoded, cleaned, computed, and missing values were coded.

### Statistical analysis

The extracted data was analyzed using the STATA version 14 statistical software. To account for differences in stratum selection and nonresponse probabilities, the data were weighted. Descriptive and inferential statistical methods were employed to present the data.

A multi-level-logistic regression model was used to assess the association between the independent variables with the dependent variable. Variables with p-value <0.05 at 95% CI with AOR were declared as statistically significant using the melogit function. Multicollinearity among the independent variables was checked using the variance inflation test (a default function for STATA 14.0 version). The final model was selected by comparing model 0, model I, model II, and model III. In Model 0 the dependent variable was analyzed with the regional variable, meaning the initial analysis focused on how the dependent variable is influenced by the regional variable alone. In Model I the effect of an individual-level variable on a dependent variable was analyzed, meaning the analysis looked at how a specific variable at an individual level impacts the dependent variable. Model II examined the dependent variable and the community variable, which is residency. Model III level examined significant variables in both model I and model II using the stepwise analysis method. The model with the least AIC and BIC was used to identify model fitness.

### Ethical considerations

Permission to access the data set from https://www.dhsprogram.com/data/available-datasets.cfm was obtained from www.DHSmeasure.org by submitting abstract. The DHS public-use datasets are protected from identification of respondents, households, or sample communities by procedures approved by the Institution Review Board. The data files do not contain household addresses or names of individuals and were used solely for the intended objective.

## Results

### Socio-demographic characteristics

A total of 1077 children aged 12-35months, whose vaccination cards were checked at home or health facility in MEDHS 2019, were included. The mean age of mothers was 28.58 ±6.27 years ranging from 17 to 49. Similarly, the mean age of children was 22.34±6.70 months with a range of 12–36 months. About half, 549 (50.97%) of the children were male (Table 1).

## Maternal and obstetric-related factors

The study revealed that almost all, 1056(98.05%) births were singleton. Concerning ANC follow-up, a bit higher than half, 504(55.96%) of mothers had 4 or more ANC visits. The study revealed that 745(69.17%) of mothers delivered in health facilities, while about two-thirds 456 (66.47%) of mothers had PNC follow (Table 2).

## Media exposure

The study revealed that, more than half, 582 (54.04%) of the children's mothers had no access to media exposure (Fig 1).

## Prevalence of age-appropriate vaccination

The prevalence of age-appropriate vaccination was 21.17% (95%CI: 18.73–23.61). Furthermore, the results of the study showed that the majority of children were vaccinated for all basic vaccinations as presented in the findings (Table 3).

## Factors associated with age-appropriate vaccination among children aged 12-35months

In the multilevel logistic regression analysis, the age of mothers, educational status of the mothers, and place of residence were statistically significant determinants of age-appropriate vaccination

The study identified that children's mothers aged > = 40 years were 4.05 times (AOR = 4.05 at 95%CI: 1.03, 15.83) more likely to vaccinate their child at an appropriate age compared to those aged between 15–19 years old. Similarly, 35-39-year-old mothers were 4.62 times (AOR = 4.62 at 95%CI: 1.27, 16.71) more likely to vaccinate their child at an appropriate age compared to those aged between 15–19 years old. In addition, 25-29-year-old mothers were 4.07 times (AOR = 4.07 at 95%CI: 1.18, 14.03) more likely to vaccinate their child at appropriate age compared to those aged between 15–19 years old.

Regarding maternal education, mothers who completed secondary education were 1.85 times (AOR = 1.85 at 95% CI: 1.06, 3.22) more likely to vaccinate their child at an appropriate age compared to those who had no education. In addition, mothers who completed primary education were 1.60 times (AOR = 1.60, 95%CI: 1.07, 2.41) more likely vaccinated at an appropriate age compared to those who had no education.

Regarding place of residence, mothers who live in rural areas were 66% (AOR = 0.34 at 95% CI: 0.23, 0.51) less likely to vaccinate their child at an appropriate age compared to those who live in urban areas (Table 4).

**Model fit statistics.** As shown in Table 4 below, Model III had the smallest AIC and BIC values, indicating that it explained the factors better than the Null Model, Model I and Model II. Therefore, this made the final model (Model III) as the best-fitted model than others.

## Discussion

The present study aimed to assess the magnitude and factors associated with age-appropriate vaccination among children aged 12–35 months in Ethiopia. Our results showed that the prevalence of age-appropriate vaccination in this study was 21.17% ((95%CI: 18.73–23.61). Significant predictors of age-appropriate vaccinations include, mother's age, educational status, and residence.

According to the findings, age-appropriate vaccination is very low (21.17%) compared to the crude prevalence of child vaccination in Ethiopia. This finding is not supported by

**Table 1. Distribution of socio-demographic characteristics (n = 1077).**

| Variables | Categories | Frequency | Percent |
|---|---|---|---|
| Region | Tigray | 134 | 12.44 |
| | Afar | 52 | 4.83 |
| | Amhara | 117 | 10.86 |
| | Oromia | 135 | 12.53 |
| | Somali | 51 | 4.74 |
| | Benishangul Gumuz | 114 | 10.58 |
| | SNNPR | 94 | 8.73 |
| | Gambela | 77 | 7.15 |
| | Harari | 85 | 7.89 |
| | Addis Ababa | 110 | 10.21 |
| | Diredawa | 108 | 10.03 |
| Place of residence | Urban | 357 | 33.15 |
| | Rural | 720 | 66.85 |
| Age of the mothers | 15–19 | 39 | 3.65 |
| | 20–24 | 218 | 20.24 |
| | 25–29 | 390 | 36.21 |
| | 30–34 | 218 | 20.24 |
| | 35–39 | 135 | 12.53 |
| | > = 40 | 77 | 7.15 |
| Mother educational status | No education | 464 | 43.08 |
| | Primary | 399 | 37.05 |
| | Secondary | 120 | 11.14 |
| | Higher | 94 | 8.73 |
| **Variable** | **Category** | **Frequency** | **Percent** |
| Religion | Orthodox | 418 | 38.81 |
| | Protestant | 181 | 16.81 |
| | Muslim | 463 | 42.99 |
| | Other * | 15 | 1.39 |
| Sex of HH++ head | Male | 888 | 82.45 |
| | Female | 189 | 17.55 |
| Wealthy status | Poor | 389 | 36.12 |
| | Middle | 167 | 15.51 |
| | Rich | 521 | 48.38 |
| Marital status | currently in union/living with a man | 1025 | 95.17 |
| | Other ** | 22 | 4.83 |
| Family size | Less than five | 376 | 34.91 |
| | Above 5 | 701 | 65.09 |
| Number of <5 children at home | < = 1 | 532 | 49.40 |
| | > = 2 | 545 | 50.60 |
| Sex of children | Male | 549 | 50.97 |
| | Female | 528 | 49.03 |
| Age of children | Between 12–24 months | 645 | 59.89 |
| | Above 24 months | 432 | 40.11 |

* = Catholic, Traditional

** = Single, widowed, Divorced

++ = Head of Household

**Table 2. Distribution of maternal and obstetric characteristics (n = 1077).**

| Variable | Category | Frequency | Percent |
|---|---|---|---|
| Place of delivery | Health facility delivery | 745 | 69.17 |
| | Home delivery | 332 | 30.83 |
| Birth order of the child | First | 270 | 25.07 |
| | 2–4 | 492 | 45.68 |
| | > = 5 | 315 | 29.25 |
| Birth in the past 5 years | One | 637 | 59.15 |
| | Two | 384 | 35.65 |
| | Three or four | 56 | 5.20 |
| Type of birth | Single birth | 1056 | 98.05 |
| | Multiple births | 21 | 1.95 |
| ANC visit | No ANC visits | 117 | 12.12 |
| | Less than 4 ANC visits | 308 | 31.92 |
| | > = 4 ANC visits | 504 | 55.96 |
| PNC | No | 230 | 33.53 |
| | Yes | 456 | 66.47 |

NB: ANC = Antenatal Care, PNC = Post Natal Care

comparable literature where this figure is very low. The majorities of the studies on this area are small samples and addressed children from limited study areas. Consequently, the age-appropriate vaccination in this study is lower than the prevalence of study findings in El

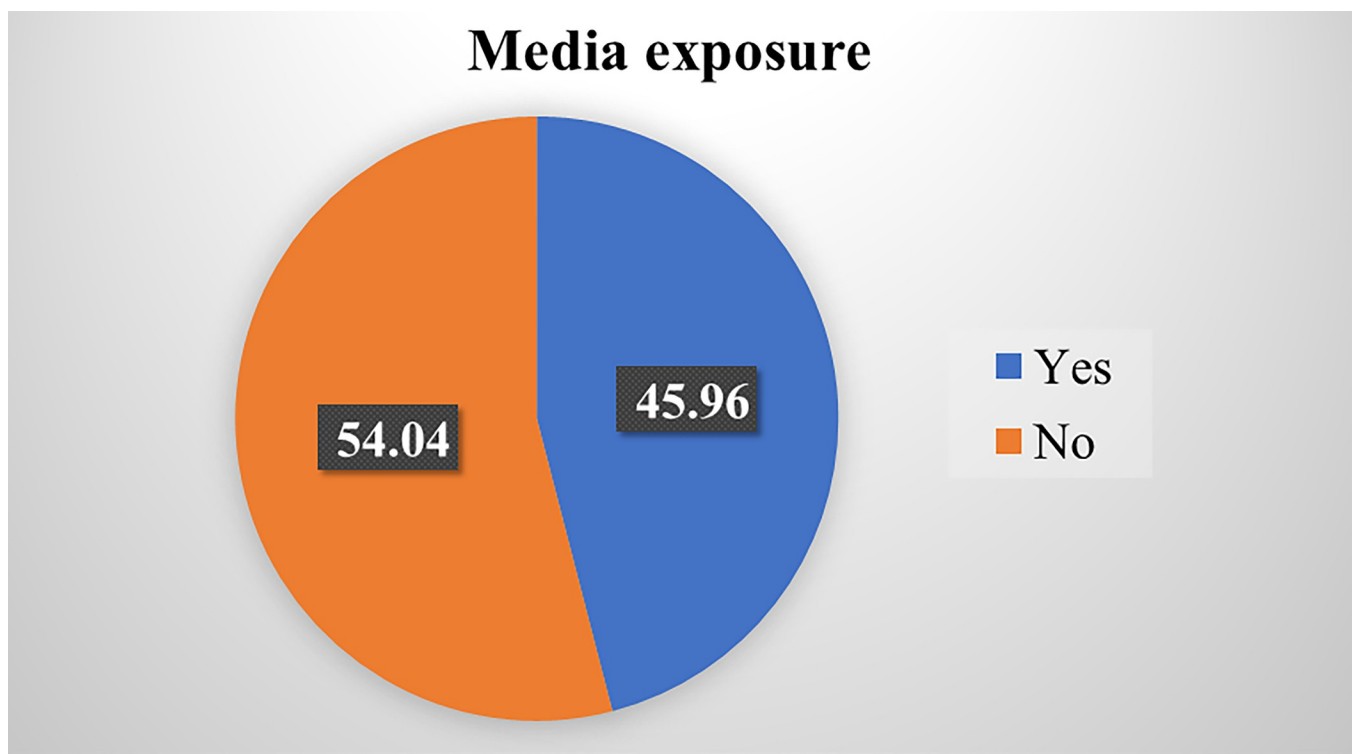

**Fig 1. Distribution of children's mothers' exposure to media.**

**Table 3. Distribution of age-appropriate vaccination among children aged 12–35 months.**

| Basic Vaccinations | Response | Appropriate time | |
|---|---|---|---|
| | | Frequency | Percent |
| BCG | Yes | 607 | 62.13 |
| | No | 370 | 37.87 |
| OPV1 | Yes | 734 | 69.71 |
| | No | 319 | 30.29 |
| Opv2 | Yes | 678 | 68.55 |
| | No | 311 | 31.45 |
| OPV3 | Yes | 662 | 71.49 |
| | No | 264 | 28.51 |
| Penta1 | Yes | 733 | 69.22 |
| | No | 326 | 30.78 |
| Penta 2 | Yes | 697 | 68.98 |
| | No | 309 | 31.02 |
| Penta 3 | Yes | 670 | 72.28 |
| | No | 257 | 27.72 |
| Measles | Yes | 473 | 60.80 |
| | No | 305 | 39.20 |

Salvador(26.7%) [27]. Similarly, this finding is lesser than the timely vaccination of specific vaccine types in India where the coverage is 31%, and 34% for BCG and MCV [28]. Generally, the very low finding of age-appropriate vaccination could be related to low economic status, low education, and other factors in Ethiopia [24, 29, 30]. This low rate of age-appropriate vaccination has serious implications like: 1) Increased disease risk: Children are more susceptible to preventable diseases, leading to higher morbidity and mortality rates. 2) Public Health Impact: Low vaccination can lead to outbreaks of VPD, straining health care resources. 3) Economic burden: Increased health care costs for treating preventable diseases can burden families and the health care system. 4) Developmental delays: Illness from VPD can causes long term health issues, affecting children's growth and development.

In the present study, children of mothers aged 25–29, 35–39, and 40 and above were more likely to vaccinate at an appropriate age compared to the children of mothers 15–19 years. This indicates that older mothers are at a better to vaccinate their children in Ethiopia. This result is supported by the study conducted in China [31], and in Tanzania where younger mothers are at a risk of low age-appropriate vaccination [32]. This could be because older mothers have valuable experience in child care, having likely raised children before, and might have more exposure to health facilities and health information [33, 34]. On the other hand, other study indicated that mothers aged above 35 years were less likely to vaccinate their children on time [35]. The difference might be due to health and mobility issue, where older mother may face health or mobility problem, and socioeconomic problems, where difference in socio economic status and access to health services can vary significantly between regions and populations. Thus, scientific research established that both extremities mother's age affects child care [34], with very young and older mothers facing unique challenges. By considering these factors we can understand that different studies may yield contrasting findings.

Moreover, children of more educated mothers were more likely vaccinated at appropriate age. Children of mothers with primary, secondary, and higher education had twice higher the odds of being vaccinated compared to children of mothers who did not attend formal education. This finding is supported by studies from China [31], Saudi Arabia [36], Tanzania [32], Cameron [37], and Ethiopia [19]. The significance among educated mothers may be due to

**Table 4. Multilevel logistic regression analysis of factors associated with age-appropriate vaccination among children aged 12–35 months.**

| Variables | Category | Model0 | Model I | Model II | Model III |
|---|---|---|---|---|---|
| Age of mother | 15–19 | - | 1 | - | 1 |
| | 20–24 | - | 1.94(.55,6.82) | - | 2.01(.56,7.11) |
| | 25–29 | - | 3.99(1.16,13.65) | - | 4.07(1.18,14.03)* |
| | 30–34 | - | 2.84(.80,10.0) | - | 2.71(.76,9.63) |
| | 35–39 | - | 4.87(1.36,17.47) | - | 4.62(1.27,16.71)* |
| | > = 40 | - | 3.96(1.02,15.31) | - | 4.05(1.03,15.83)* |
| Place of delivery | Health facility | - | 1 | - | 1 |
| | Home/other | - | .53(.32,82) | - | .63(.40,.98) |
| Media Exposure | Yes | - | 1.46(1.00,2.15) | - | 1.01(.68,1.52) |
| | No | - | 1 | - | 1 |
| Educational status | No education | - | 1 | - | 1 |
| | Primary | - | 1.73(1.15,2.59) | - | 1.60(1.07,2.41)* |
| | Secondary | - | 2.16(1.25,3.74) | - | 1.85(1.06,3.22)* |
| | Higher | - | 2.04(1.13,3.69) | - | 1.59(.88,2.88) |
| Child birth in the past 5 years | One | - | 1 | - | 1 |
| | Two | - | .80(.56,.1.13) | - | .84(.59,1.19) |
| | Three/Four | - | .76(.32,1.82) | - | .76(.32,1.84) |
| Religion | Orthodox | - | 1 | - | 1 |
| | Protestant | - | .62(.35,1.11) | - | .62(.38,1.02) |
| | Muslim | - | .81(.55,1.19) | - | .79(.55,1.12) |
| | Other | - | 1.21(.29,4.91) | - | 1.10(.28,4.23) |
| Residence | Urban | - | | 1 | 1 |
| | Rural | - | | .26(.18,.37) | .34(.23,.51)* |
| Std err | | .221623 | 0.8586 | 0.0883094 | 3.15e-16 |
| Log-likelihood | | -539.082 | -508.2364 | -512.7568 | -494.962 |
| AIC | | 1082.16 | 1050.473 | 1031.514 | 1023.924 |
| BIC | | 1092.12 | 1135.166 | 1046.459 | 1108.617 |

\* = statistically significant at P<0.05, 1 = reference category, AIC = Akaike information criterion, BIC = Bayesian Information Criterion

their awareness of the benefits of child vaccination. In addition, educated mothers are often better at understanding the health care providers' counseling and messages on vaccination during awareness creation sessions at health facilities and at the community level. Many mothers receive childcare information from health extension workers and women development armies in Ethiopia [38, 39]. In support of this, scientific research has confirmed that children of educated mothers have a higher rate of complete immunization due to better literacy skills and health-seeking behavior, which in turn improves immunization uptake for their children [40, 41].

Furthermore, children of mothers who live in rural areas were less likely vaccinated at the appropriate age compared to those who live in rural areas. This finding is supported by other studies in Nigeria [42] and Kenya [43]. This could be due to various factors like limited access to health care services, inadequate infrastructure, and lower educational levels in rural settings. This implies that the government has to look into another strategic direction like community engagement and targeted intervention mechanisms to minimize rural-urban disparities.

Despite the current analysis being based on the most recent nationally representative survey data and using a multilevel logistic regression to handle the variation within the level due to the hierarchical nature of the EMDHS data, it is not free of limitation. One of the limitations

of the study is the difficulty of showing temporal association because of the cross-sectional nature of the study. Another limitation is the potential recall bias, EMDHS survey is based on the participants' reports,

## Conclusion

The study concluded that the prevalence of age-appropriate vaccination of children in Ethiopia is significantly below the desired level. Factors such as mother's age, education, and residence were significant predictors of age-appropriate vaccinations. Hence, the stakeholders should prioritize timely vaccination of children, similar to as the efforts made to increase overall coverage.

## Acknowledgments

The authors would like to express gratitude to www.DHSmeasure.org for granting access to the datasets.

## Author Contributions

**Conceptualization:** Bekelu Teka Worku, Eshetu Alemayehu Wordofa, Gadisa Senbeto, Ebissa Bayana Kebede, Fira Abamecha, Gurmessa Tura Debela, Negalign Birhanu, Yibeltal Siraneh, Dessalegn Tamiru.

**Data curation:** Bekelu Teka Worku, Eshetu Alemayehu Wordofa, Beakal Zinab, Ebissa Bayana Kebede, Gurmessa Tura Debela, Negalign Birhanu, Yibeltal Siraneh, Dessalegn Tamiru.

**Formal analysis:** Eshetu Alemayehu Wordofa.

**Funding acquisition:** Eshetu Alemayehu Wordofa.

**Investigation:** Bekelu Teka Worku, Beakal Zinab, Ebissa Bayana Kebede, Fira Abamecha, Yibeltal Siraneh, Dessalegn Tamiru.

**Methodology:** Bekelu Teka Worku, Eshetu Alemayehu Wordofa, Gadisa Senbeto, Ebissa Bayana Kebede.

**Project administration:** Bekelu Teka Worku, Gadisa Senbeto, Beakal Zinab, Fira Abamecha, Gurmessa Tura Debela, Negalign Birhanu, Dessalegn Tamiru.

**Resources:** Bekelu Teka Worku, Gadisa Senbeto, Beakal Zinab, Negalign Birhanu, Yibeltal Siraneh.

**Software:** Eshetu Alemayehu Wordofa.

**Supervision:** Gadisa Senbeto, Beakal Zinab, Fira Abamecha, Gurmessa Tura Debela, Negalign Birhanu, Yibeltal Siraneh, Dessalegn Tamiru.

**Validation:** Eshetu Alemayehu Wordofa, Gadisa Senbeto, Fira Abamecha, Gurmessa Tura Debela, Negalign Birhanu, Yibeltal Siraneh, Dessalegn Tamiru.

**Visualization:** Eshetu Alemayehu Wordofa, Gadisa Senbeto, Beakal Zinab, Ebissa Bayana Kebede, Fira Abamecha, Gurmessa Tura Debela, Negalign Birhanu, Dessalegn Tamiru.

**Writing – original draft:** Bekelu Teka Worku, Eshetu Alemayehu Wordofa, Ebissa Bayana Kebede.

**Writing – review & editing:** Bekelu Teka Worku, Eshetu Alemayehu Wordofa, Gadisa Senbeto, Beakal Zinab, Ebissa Bayana Kebede, Fira Abamecha, Gurmessa Tura Debela, Negalign Birhanu, Yibeltal Siraneh, Dessalegn Tamiru.

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
