## [Decision Letter · Decision Letter 0]

19 Aug 2024

PONE-D-24-22259Age-appropriate Vaccination and Associated Factors among Children Aged 12- 35 Months in Ethiopia: A Multi-Level AnalysisPLOS ONE

Dear Dr. Bayana,

Thank you for submitting your manuscript to PLOS ONE. After careful consideration, we feel that it has merit but does not fully meet PLOS ONE’s publication criteria as it currently stands. Therefore, we invite you to submit a revised version of the manuscript that addresses the points raised during the review process.

We look forward to receiving your revised manuscript.

Kind regards,

Kahsu Gebrekidan, Ph.D.

Academic Editor

PLOS ONE

Journal Requirements:

Reviewers' comments:

Reviewer's Responses to Questions

**Comments to the Author**

1. Is the manuscript technically sound, and do the data support the conclusions?

Reviewer #1: No

Reviewer #2: Yes

2. Has the statistical analysis been performed appropriately and rigorously? 

Reviewer #1: I Don't Know

Reviewer #2: Yes

3. Have the authors made all data underlying the findings in their manuscript fully available?

Reviewer #1: No

Reviewer #2: Yes

4. Is the manuscript presented in an intelligible fashion and written in standard English?

Reviewer #1: No

Reviewer #2: No

5. Review Comments to the Author

Reviewer #1: Results (Statistics)

I would recommend that the statistical method and calculations be reviewed for appropriateness and accuracy by a statistician. Though I have the following comments:

Page 15 & 22 & Table 4: The text discusses statistical models zero (0) through three (III). These are also displayed in Table 4. However, the text on page 22 states: “As shown in Table 4 below, a small number of AIC and BIC was Model IV, which indicates...” I am unable to find other references to Model IV.

Population size is reported as 1,077 children from ages 12 to 35 months. Table 3 (pg 20) provides a breakdown of vaccines. It is unclear if this reflects full immunization coverage or age-appropriate coverage. The text reads: “Furthermore, theresults of the study showed that the majority of children were vaccinated for all basic vaccinations as presented in Table 3.” Yet the table heading/description reads “age-appropriate” vaccination. This is confusing. It is also unclear, if “majority” of children are vaccinated for all basic vaccinations, why the authors report the age-appropriate vaccination as 21%.

The total “No of ANC visits” (929) and PNC follow-up contacts (686) do not total 1,077 children enrolled into this study. This difference needs to be explained.

Furthermore, the totals for “Yes” and “No” do not match expected totals for the reported population size. With 1,077 babies aged 12 months or above the expected total should be 1,077 for BCG, OPV1-3 and Penta 1-3 and Measles. What is reported is: 977 (BCG), 1053 (OPV1), 989 (OPV2), 926 (OPV3), 1059 (Penta1), 1006 (Penta2), 927 (Penta3) and 778 (Measles) respectively. Furthermore, assuming that each child received OPV and Penta at the same time (pg13) the totals for OPV1/Penta1, OPV2/Penta2, OPV3/Penta3 should be the same, which they are not. These discrepancies may confuse the reader.

Similarly, the percentage for the “appropriate time frequency” is reported to be above 60% for each vaccine. It is unclear how the authors derive at a prevalence of 21.7% as reported in the text.

On page 21 the authors report odds ratios for associated factors (4.05, 4.62, 4.07). It is not clear where this information is presented in previous tables and how these AORs were calculated. Childbirth in the past five years, maternal age and educational level were ported as significant, but no p-Value is reported in any of the tables.

On page 21 the authors report decrease coverage: “...areas were 66% (AOR=.34 at 95%CI.23,.51) less likely...”. In keeping with the previous AORs, could this percentage be reported as an AOR?

It may be possible that the above questions arise due to lack of understanding (on statistical methods used) on my part. Alternatively, the authors may need to clarify their analysis steps and address the differences between the totals in Table 3, the total population size of 1,077 as well as the 60% and 21% difference.

Major concerns:

Perhaps the authors could differentiate between full coverage (or complete overage) and age-appropriate vaccine coverage. It may be important to understand the consequences of delayed vaccination and why the international targets report (full) coverage as opposed to age-appropriate coverage. This is of importance as the authors state “...ensuring universal vaccination through routine and catch-up vaccination schedules is an essential part of quality healthcare in nations. It is linked to improved health outcomes, cognitive development, productivity, and cost savings (10-12).” (pg 9). This last statement seems to invalidate the authors’ concern for lack of age-appropriate vaccination, and thus also their study.

Page 10 states: “pooled full vaccination coverage in Ethiopia is greater than 65% which is higher than all these figures indicating poor age-appropriate vaccination in different regions...” This statement refers to various literature reporting age-appropriate vaccine coverage, implying the authors might be comparing two different measures. This further highlights the need for clarification of these two terms.

The results section states three factors of significance (age of mothers, educational status, childbirth in the last five years) (pg 21), but the discussion omits the factor of “childbirth in the last five years’ and adds “place of residence” instead. It may be important to elaborate if “Place of residence” (urban vs rural) refers to distance to nearest facility, or level of facility or both for the rural population. Moreover, the discussion highlights “economic status” as a factor but is not placed in context of what is reported for this study. The omission of childbirth in the discussion and the addition of place of residence and economics may need an explanation. Similarly, the abstract needs to be aligned accordingly.

The authors acknowledge (pg10): “some studies have assessed age-appropriate vaccination in Ethiopia, no study used the most recent national data to evaluate this issue...” It is unclear why this study needs to repeat the work of previous studies. The methodology and findings of these previous studies are not explained in the text. The authors may want to address what they were expecting to find in this study that might have been different from the previous studies.

The authors report strategies implemented in Ethiopia to improve overall vaccination coverage, but do not report what they targeted, what they achieved and what the remaining gaps are. Similarly, the discussion does not delve into the health implications for under-vaccination. Perhaps this could be elaborated more.

The conclusion recommends “due attention”. It is unclear what actions are expected. The authors may wish to expand what was done elsewhere, which interventions failed and were successful and which may be implementable in Ethiopia. This is a gap that may be covered with reference to additional review of literature on similar topics.

This study found: “Similarly, 35-39-year-old mothers were 4.62 times...” more likely to have their children immunized. In the discussion authors report an opposite finding: “On the other hand, mothers aged above 35 years were less likely to vaccinate their children on time (38).” This seems to be an important contrast and could be explicated further.

Reviewer #2: The manuscript addresses an important global challenge in Paediatrics and Child Health. The authors have used secondary data from an existing country-specific database. The statistical analysis is adequate. The findings are well described.

The manuscript will benefit a lot from copyediting to improve the language used in the manuscript. Just by way of example where the language can be improved:

Introduction: Last sentence on page 3 - However, the burden of VPD remains not easy in today's world - could be phrased better - However, the burden of VPD remains high OR a challenge? in today's world

Introduction, paragraph 1 last sentence: Does not read well because it starts with the words: On the other way.......

Materials and Methods: The opening paragraph on Ethiopia being a populous country, does not read well

Result (Heading): should be Results (Plural)

6. PLOS authors have the option to publish the peer review history of their article (what does this mean?). If published, this will include your full peer review and any attached files.

Reviewer #1: **Yes: **Beatrix Callard

Reviewer #2: **Yes: **Mawela Mothahadini

---

## [Author Response · Author response to Decision Letter 0]

23 Aug 2024

Reviewer #1: Results (Statistics)

I would recommend that the statistical method and calculations be reviewed for appropriateness and accuracy by a statistician. Though I have the following comments:

Thank you, dear reviewer, for your concern. We have tried to address your comments. One of our authors is statistician.

Page 15 & 22 & Table 4: The text discusses statistical models zero (0) through three (III). These are also displayed in Table 4. However, the text on page 22 states: “As shown in Table 4 below, a small number of AIC and BIC was Model IV, which indicates...” I am unable to find other references to Model IV.

Thank you for your feedback. 

We revised now; it was editorial problem and has been fixed as per your suggestions. Actually it is not model IV; it is model III

Population size is reported as 1,077 children from ages 12 to 35 months. Table 3 (pg 20) provides a breakdown of vaccines. It is unclear if this reflects full immunization coverage or age-appropriate coverage. The text reads: “Furthermore, the results of the study showed that the majority of children were vaccinated for all basic vaccinations as presented in Table 3.” Yet the table heading/description reads “age-appropriate” vaccination. This is confusing. It is also unclear, if “majority” of children are vaccinated for all basic vaccinations, why the authors report the age-appropriate vaccination as 21%.

Thank you for your concern

We understand your fear; however, the table title is not about immunization coverage. It is about age-appropriate vaccination. The other thing is that being vaccinated for basic vaccination may not reflect that the children were vaccinated at appropriate age. When we talk about vaccination coverage, we are talking about all recommended vaccination regardless of timing. When we say age-appropriate, we mean that children received vaccines at the recommended age. Therefore, when we say the majority were vaccinated for all vaccinations, it indicates full immunization coverage. When we say 21.7% received it at appropriate age, we mean that only 21.7% received it within recommended age.

The total “No of ANC visits” (929) and PNC follow-up contacts (686) do not total 1,077 children enrolled into this study. This difference needs to be explained.

Thank you for your suggestions. 

Obviously, ANC means Antenatal care, and PNC is post-natal care. Hence, they are different conceptually, and we do not add them together. Of samples included, the 929 has ANC follow-up and 686 had post-natal care follow up at time of the survey. Even though all pregnant women are expected to follow ANC services, some pregnant women may not follow because of different reasons. Similarly, after delivery, mothers are expected follow PNC services but many of the miss it because of different reason again. 

Furthermore, the totals for “Yes” and “No” do not match expected totals for the reported population size. With 1,077 babies aged 12 months or above the expected total should be 1,077 for BCG, OPV1-3 and Penta 1-3 and Measles. What is reported is: 977 (BCG), 1053 (OPV1), 989 (OPV2), 926 (OPV3), 1059 (Penta1), 1006 (Penta2), 927 (Penta3) and 778 (Measles) respectively. 

Great! 

Yes, the discrepancy can be justified. The eligible population for this study were 1077. But there some children who were not vaccinated, or may have been vaccinated but not confirmed because absence of Immunization card at home or health institution during the survey. For example, out 1077 eligible children, 977 were vaccinated for BCG, which indicates 100 children were either not vaccinated for BCG or it was not verified at time of data collection (i.e., if there is no Immunization card at home or at the health facility, with oral report vaccination status was not used as verification) . 

The other thing is that the “Yes” and “No” in the table indicate whether children were vaccinated. Thus, if child was vaccinated at appropriate age, it was counted as “Yes”, otherwise, “NO.”

Furthermore, assuming that each child received OPV and Penta at the same time (pg13) the totals for OPV1/Penta1, OPV2/Penta2, OPV3/Penta3 should be the same, which they are not. These discrepancies may confuse the reader.

Thank you for your comments. 

Conceptually you are right. However, in reality there a time when either the vaccine may stock out, leading to differences in the totals. Additionally some children may miss one of the vaccines because of various reasons like illness or parental preferences. So it may not be the same.

Similarly, the percentage for the “appropriate time frequency” is reported to be above 60% for each vaccine. It is unclear how the authors derive at a prevalence of 21.7% as reported in the text.

Thank you for your feedback. 

The discrepancy between appropriate time frequency and the prevalence of 21.7% can be explained as follows:

• Appropriate time frequency: This indicates the proportion of children who received each vaccine at the recommended age. For example, for BCG out of 1077 samples, 977 children were vaccinated where 607 received at appropriate time, and 370 received not received at appropriate time. So we can calculate its proportion as (670/977)*100 and (370/977)*100. So for others also we can proceed in similar patterns.

• Prevalence of 21.7%: This figure refers to the overall prevalence of children who were vaccinated at appropriate age. It is calculated by using World health Organization Window period. Thus, the calculation method is well explained and available under operational definition in the method section. 

On page 21 the authors report odds ratios for associated factors (4.05, 4.62, 4.07). It is not clear where this information is presented in previous tables and how these AORs were calculated. Childbirth in the past five years, maternal age and educational level were ported as significant, but no p-Value is reported in any of the tables.

Thank you for the concerns 

You find the full information in table 4. P value is indicated by using asterisk (*). 

On page 21 the authors report decrease coverage: “...areas were 66% (AOR=.34 at 95%CI.23,.51) less likely...”. In keeping with the previous AORs, could this percentage be reported as an AOR?

Thank you for the queries

The AOR is 0.34, so if the AOR is less than 1, we have to change it percentage for interpretation. Hence AOR is 0.34 at 95%CI, (1-0.34)*100 =66%. Thus, for interpretation it is better if we use in this way.

It may be possible that the above questions arise due to lack of understanding (on statistical methods used) on my part. Alternatively, the authors may need to clarify their analysis steps and address the differences between the totals in Table 3, the total population size of 1,077 as well as the 60% and 21% difference.

Thank you very much for your concerns. Please check the responses provided in each comment you raised.

Major concerns:

Perhaps the authors could differentiate between full coverage (or complete overage) and age-appropriate vaccine coverage. It may be important to understand the consequences of delayed vaccination and why the international targets report (full) coverage as opposed to age-appropriate coverage. This is of importance as the authors state “...ensuring universal vaccination through routine and catch-up vaccination schedules is an essential part of quality healthcare in nations. It is linked to improved health outcomes, cognitive development, productivity, and cost savings (10-12).” (pg 9). This last statement seems to invalidate the authors’ concern for lack of age-appropriate vaccination, and thus also their study.

Thank you for the concerns.

It doesn’t invalidate; rather, it validates. If the child cannot be vaccinated at appropriate age, the child is at risk for many health-related problems. So it does not invalidate. Regarding age appropriate and full coverage, we included in aforementioned responses. 

Page 10 states: “pooled full vaccination coverage in Ethiopia is greater than 65% which is higher than all these figures indicating poor age-appropriate vaccination in different regions...” This statement refers to various literature reporting age-appropriate vaccine coverage, implying the authors might be comparing two different measures. This further highlights the need for clarification of these two terms.

Thank you.

The first one talk about coverage and the concept is related with difference in regions in Ethiopia. During the survey, there were nine regions and two administrative cities. There are differences of vaccination coverage across the regions, however it was less than 50% but the pooled is more than the specific regions, which accounted for almost 65%. Please find the detailed description under introduction in the 4th paragraph

The results section states three factors of significance (age of mothers, educational status, childbirth in the last five years) (pg 21), but the discussion omits the factor of “childbirth in the last five years’ and adds “place of residence” instead. It may be important to elaborate if “Place of residence” (urban vs rural) refers to distance to nearest facility, or level of facility or both for the rural population. Moreover, the discussion highlights “economic status” as a factor but is not placed in context of what is reported for this study. The omission of childbirth in the discussion and the addition of place of residence and economics may need an explanation. Similarly, the abstract needs to be aligned accordingly.

Thank you

We corrected now. There was an editorial problem. Thus, it is place of residence which was statistically significant not childbirth in the past five years. So we corrected in result section.

When we say place residence, we mean a place where the participants are living, either in urban or rural areas. It is all about geographical location not near or far in relation to health facilities 

The authors acknowledge (pg10): “some studies have assessed age-appropriate vaccination in Ethiopia, no study used the most recent national data to evaluate this issue...” It is unclear why this study needs to repeat the work of previous studies. The methodology and findings of these previous studies are not explained in the text. The authors may want to address what they were expecting to find in this study that might have been different from the previous studies.

Thank you

Yes, as it is already explained in the paragraph, 1) the study conducted are not nationally representative but conducted in specific location. 2) None of the studies have been conducted from the recent nationally representative data. The National data is very important and representative of children in the country.

The authors report strategies implemented in Ethiopia to improve overall vaccination coverage, but do not report what they targeted, what they achieved and what the remaining gaps are. Similarly, the discussion does not delve into the health implications for under-vaccination. Perhaps this could be elaborated more.

Thank you for your concerns.

Regarding the strategies, it is already available under introduction part, which you may find in paragraph five in detail. 

Regarding the health implications, we included it now in the discussion as follows:

This low rate of age-appropriate vaccination has serious implications like: 1) Increased disease risk: Children are more susceptible to preventable diseases, leading to higher morbidity and mortality rates. 2) Public Health Impact: Low vaccination can lead to outbreaks of VPD, straining health care resources. 3) Economic burden: Increased health care costs for treating preventable diseases can burden families and the health care system. 4) Developmental delays: Illness from VPD can causes long term health issues, affecting children’s growth and development.

The conclusion recommends “due attention”. It is unclear what actions are expected. The authors may wish to expand what was done elsewhere, which interventions failed and were successful and which may be implementable in Ethiopia. This is a gap that may be covered with reference to additional review of literature on similar topics.

Thank you for your reflection: We corrected now

 Hence, the stakeholders should prioritize timely vaccination of children, similar to as the efforts made to increase overall coverage. 

This study found: “Similarly, 35-39-year-old mothers were 4.62 times...” more likely to have their children immunized. In the discussion authors report an opposite finding: “On the other hand, mothers aged above 35 years were less likely to vaccinate their children on time (38).” This seems to be an important contrast and could be explicated further.

Thank you for your concern, we tried to explain as follows which is also included in the manuscript

 The difference might be due to health and mobility issue, where older mother may face health or mobility problem, and socioeconomic problems, where difference in socio economic status and access to health services can vary significantly between regions and populations. Thus, scientific research established that both extremities mother's age affects child care (37), with very young and older mothers facing unique challenges. By considering these factors we can understand that different studies may yield contrasting findings.

Reviewer #2: The manuscript addresses an important global challenge in Paediatrics and Child Health. The authors have used secondary data from an existing country-specific database. The statistical analysis is adequate. The findings are well described.

Dear reviewer thank you for your encouragement.

The manuscript will benefit a lot from copyediting to improve the language used in the manuscript. Just by way of example where the language can be improved:

Thank you for your feedback.

We tried to our best to carefully correct all typos, grammar and spelling in the revised version. 

Introduction: Last sentence on page 3 - However, the burden of VPD remains not easy in today's world - could be phrased better - However, the burden of VPD remains high OR a challenge? in today's world

Thank you for the comment. We corrected the whole sentences as the follows: 

Despite these efforts, the burden of vaccine-preventable disease remains a global challenge. 

Introduction, paragraph 1 last sentence: Does not read well because it starts with the words: On the other way.......

Thank you for the comments: We corrected as follows: Additionally, the magnitude of missed…

Materials and Methods: The opening paragraph on Ethiopia being a populous country, does not read well

Thank you for your concern. We edited to make more meaningful 

The study was conducted in Ethiopia, the largest nation in the Horn of Africa. Ethiopia is the second most populous country in Africa and ranks 11th in the world populous country

Result (Heading): should be Results (Plural)

Thank you

We have revised as per your suggestions as follows:

Results

---

## [Decision Letter · Decision Letter 1]

16 Sep 2024

PONE-D-24-22259R1Age-appropriate Vaccination and Associated Factors among Children Aged 12- 35 Months in Ethiopia: A Multi-Level AnalysisPLOS ONE

Dear Dr. Bayana,

Thank you for submitting your manuscript to PLOS ONE. After careful consideration, we feel that it has merit but does not fully meet PLOS ONE’s publication criteria as it currently stands. Therefore, we invite you to submit a revised version of the manuscript that addresses the points raised during the review process.

Please submit your revised manuscript by Oct 31 2024 11:59PM. If you will need more time than this to complete your revisions, please reply to this message or contact the journal office at plosone@plos.org. Please include the following items when submitting your revised manuscript:A rebuttal letter that responds to each point raised by the academic editor and reviewer(s). You should upload this letter as a separate file labeled 'Response to Reviewers'.A marked-up copy of your manuscript that highlights changes made to the original version. You should upload this as a separate file labeled 'Revised Manuscript with Track Changes'.An unmarked version of your revised paper without tracked changes. You should upload this as a separate file labeled 'Manuscript'.If applicable, we recommend that you deposit your laboratory protocols in protocols.io to enhance the reproducibility of your results. Protocols.io assigns your protocol its own identifier (DOI) so that it can be cited independently in the future. For instructions see: https://journals.plos.org/plosone/s/submission-guidelines#loc-laboratory-protocols. Additionally, PLOS ONE offers an option for publishing peer-reviewed Lab Protocol articles, which describe protocols hosted on protocols.io. Read more information on sharing protocols at https://plos.org/protocols?utm_medium=editorial-email&utm_source=authorletters&utm_campaign=protocols.

We look forward to receiving your revised manuscript.

Kind regards,

Kahsu Gebrekidan, Ph.D.

Academic Editor

PLOS ONE

Journal Requirements:

Reviewers' comments:

Reviewer's Responses to Questions

**Comments to the Author**

1. If the authors have adequately addressed your comments raised in a previous round of review and you feel that this manuscript is now acceptable for publication, you may indicate that here to bypass the “Comments to the Author” section, enter your conflict of interest statement in the “Confidential to Editor” section, and submit your "Accept" recommendation.

Reviewer #1: (No Response)

Reviewer #2: All comments have been addressed

2. Is the manuscript technically sound, and do the data support the conclusions?

Reviewer #1: Yes

Reviewer #2: Yes

3. Has the statistical analysis been performed appropriately and rigorously? 

Reviewer #1: Yes

Reviewer #2: Yes

4. Have the authors made all data underlying the findings in their manuscript fully available?

Reviewer #1: Yes

Reviewer #2: Yes

5. Is the manuscript presented in an intelligible fashion and written in standard English?

Reviewer #1: Yes

Reviewer #2: Yes

6. Review Comments to the Author

Reviewer #1: The authors have clarified all queries raised. If wordcount and space allow, I would like to see some of the clarifying responses reflected in the article's text as these add context and clarity. Minor comments remain:

Line/s:

13: mothers aged (remove d)

28: rotavirus and rotavirus (duplication)

38/42: vaccine-prevalent (preventable)

43: Nowadays (use different phrase - currently, at present, at time of writing)

85: Children were excluded if the dates of vaccination were inconsistent with their birth dates. (Please reword to add clarity)

33-34 &105-106 & 112, 113 - specify content of pentavalent vaccine (consistency)

148: The mean age of mothers was a 28.58 ±6.27 year (remove 'a' and add years)

155: Table 1 - abbreviation not explained - "Sex of HH++ head" (add abbreviation to bottom of table)

229: specify abbreviation earlier (e.g. line 38) (VPD - introduce abbreviation in text earlier)

Reviewer #2: The changes made have enhanced the quality of the manuscript. Just two very minor editorial errors can still be corrected;

Line 3 in the abstract: public health measure should be singular, not plural

Line 35: should be the target not a target.

7. PLOS authors have the option to publish the peer review history of their article (what does this mean?). If published, this will include your full peer review and any attached files.

Reviewer #1: **Yes: **Beatrix Callard

Reviewer #2: **Yes: **Mawela MPB

---

## [Author Response · Author response to Decision Letter 1]

20 Sep 2024

Reviewer 1

Line/s:

13: mothers aged (remove d)

Thank you reviewer for your concern, we removed letter d

28: rotavirus and rotavirus (duplication)

Thank you reviewer for your feedback, we removed the duplication

38/42: vaccine-prevalent (preventable)

Thank you, reviewer, for your concern. We repacked prevalent to preventable

43: Nowadays (use different phrase - currently, at present, at time of writing)

Thank you for your suggestions, we replaced with currently

85: Children were excluded if the dates of vaccination were inconsistent with their birth dates. (Please reword to add clarity)

Thank you for your concern, we rephrased as children were excluded if their vaccination dates didn’t align with their birth dates. 

33-34 &105-106 & 112, 113 - specify content of pentavalent vaccine (consistency)

Thank you for your suggestions. We tried to make consistent especially on line 105-106 & 112, 113 but regarding the concepts on 33-34, the concept of Penta was not initiated at that moment. 

148: The mean age of mothers was a 28.58 ±6.27 year (remove 'a' and add years)

Thank you reviewer for your concern, we removed letter a and added years

155: Table 1 - abbreviation not explained - "Sex of HH++ head" (add abbreviation to bottom of table)

Thank you for your suggestions. Now its full name is included under the table: HH = Head of Household

229: specify abbreviation earlier (e.g. line 38) (VPD - introduce abbreviation in text earlier)

Thank you for your suggestions. VPD –now it is included in text

Reviewer #2: The changes made have enhanced the quality of the manuscript. Just two very minor editorial errors can still be corrected;

Dear reviewer thank you for your inspiration.

Line 3 in the abstract: public health measure should be singular, not plural

Thank you for your concern. Now it is corrected to public health measure

Line 35: should be the target not a target.

Thank you for the suggestion. We removed a and replaced by the

---

## [Decision Letter · Decision Letter 2]

25 Sep 2024

Age-appropriate Vaccination and Associated Factors among Children Aged 12- 35 Months in Ethiopia: A Multi-Level Analysis

PONE-D-24-22259R2

Dear Mr. Ebissa,

We’re pleased to inform you that your manuscript has been judged scientifically suitable for publication and will be formally accepted for publication once it meets all outstanding technical requirements.

Kind regards,

Kahsu Gebrekidan, Ph.D.

Academic Editor

PLOS ONE

Additional Editor Comments (optional):

Reviewers' comments:

Reviewer's Responses to Questions

**Comments to the Author**

1. If the authors have adequately addressed your comments raised in a previous round of review and you feel that this manuscript is now acceptable for publication, you may indicate that here to bypass the “Comments to the Author” section, enter your conflict of interest statement in the “Confidential to Editor” section, and submit your "Accept" recommendation.

Reviewer #1: All comments have been addressed

2. Is the manuscript technically sound, and do the data support the conclusions?

Reviewer #1: Yes

3. Has the statistical analysis been performed appropriately and rigorously? 

Reviewer #1: Yes

4. Have the authors made all data underlying the findings in their manuscript fully available?

Reviewer #1: Yes

5. Is the manuscript presented in an intelligible fashion and written in standard English?

Reviewer #1: Yes

6. Review Comments to the Author

Reviewer #1: No further comments. The authors have addressed all relevant comments and made the necessary corrections.

7. PLOS authors have the option to publish the peer review history of their article (what does this mean?). If published, this will include your full peer review and any attached files.

Reviewer #1: **Yes: **Beatrix Callard

---

## [Editor Report · Acceptance letter]

2 Oct 2024

PONE-D-24-22259R2 

PLOS ONE

Dear Dr. Kebede, 

I'm pleased to inform you that your manuscript has been deemed suitable for publication in PLOS ONE. Congratulations! Your manuscript is now being handed over to our production team.

Kind regards, 

on behalf of

Dr. Kahsu Gebrekidan 

Academic Editor

PLOS ONE